# Hybrid Methodology to Improve Health Status Utility Values Derivation Using EQ-5D-5L and Advanced Multi-Criteria Techniques

**DOI:** 10.3390/ijerph17041423

**Published:** 2020-02-22

**Authors:** Johanna Vásquez, Sergio Botero

**Affiliations:** 1Departamento de Economía, Facultad de Ciencias Humanas y Económicas, Universidad Nacional de Colombia Sede Medellín, Medellín 050034, Colombia; 2Departamento de Ingeniería de la Organización, Facultad de Minas, Universidad Nacional de Colombia Sede Medellín, Medellín 050034, Colombia; sbotero@unal.edu.co

**Keywords:** MCDM, AHP, TOPSIS, elicit preferences, health utility values

## Abstract

This paper presented a new approach to the calculation of quality-adjusted life years (QALY) based on multi-criteria decision-making (MCDM) methods and using the EQ-5D-5L questionnaire. The health status utility values were calculated through a hybrid methodology. We combined the analytic hierarchy process (AHP), the AHP with a D-number extended fuzzy preference relation (D-AHP), the fuzzy analytic hierarchy process (F-AHP), and the technique for order preference by similarity to the ideal solution (TOPSIS) to obtain individual and aggregated utility values. The preference data were elicited using a sample of individuals from a Colombian university. In all tested methods, the ordinal preferences were consistent, and the weights were compared using the Euclidean distance criterion (EDC). We identified F-AHP-TOPSIS as the optimal method; its benefits were associated with modeling the response options of the EQ-5D in linguistic terms, it gave the best approximation to the initial preferences according to EDC, and it could be used as an alternative to the known prioritization method. This hybrid methodology was particularly useful in certain medical decisions concerned with understanding how a specific person values his or her current health or possible health outcomes from different interventions in small population samples and studies carried out in low- and middle-low-income countries.

## 1. Introduction

Decision-making is a natural process in which a person chooses one course of action from a finite set, where each action leads to possible states according to a probability distribution associated with their expected utility value [1]. Theoretically, in the healthcare sector, the process does not differ from the way it operates in other fields, but individual characteristics make it difficult to predict how an individual will respond to a specific treatment, program or medication, which adds complexity to the task. Generally, healthcare decision-making is based on economic assessment through a cost-utility or effectiveness analysis. In these studies, utility values are measured through social preferences for health states and are expressed in quality-adjusted life years (QALY). This approach takes expected utility theory and multi-attribute utility theory (MAUT) techniques as the methodological platform for comparing intervention alternatives and allocating resources efficiently [2,3,4,5,6,7,8,9,10]. The prioritization method to measure utility values uses the EQ-5D questionnaire and applies the time tradeoff (TTO) to elicit preferences and the visual analog scale (VAS) to establish a reference point. This conventional measurement assigns only a utility value that describes a particular health status and aggregates individual subjective utility values to make objective social decisions. Then, healthcare institutions make decisions based on social preferences obtained from a population sample, assuming several theoretical axioms that generate biased QALY weights [11,12]. However, what is actually needed in making most medical decisions is to understand how a specific person values his or her current health or possible health outcomes from interventions with small samples; the situation worsens when some countries make medical decisions taking utility value sets from other countries’ populations. The present paper explained how advancement in multi-criteria decision-making (MCDM) methods, such as Analytic Hierarchy Process (AHP), Analytic Hierarchy Process extended by D-number (D-AHP) and Fuzzy Analytic Hierarchy Process (F-AHP), offered alternative utility measure methods to assign weights to health status through individual preferences. In these methods, preferences were elicited when each person assessed the relative importance of each health dimension in the EQ-5D survey. These preferences, combined with the disability levels associated with EQ-5D answer levels, reflected individual characteristics and became the essential reference point for the utility value weights, as Bernoulli stated (1954) [13]; also, the judgments showed the experience associated with health and illness. Therefore, we proposed a hybrid methodology for calculating not only social preferences but also individual preferences. The methods used were easy-to-implement, low-cost primary data collection methods, and it was possible to assign a utility value for each declared health status. This methodology could be applied to small population samples and was very appropriate for low- and middle-low-income countries. The paper structure is as follows: Section 2 states the current issues. Section 3 describes the classical AHP method, as well as the D-AHP, F-AHP, and the Technique for Order Preference by Similarity to the Ideal Solution (TOPSIS); it also describes key concepts and the steps for calculating the criteria priority weights, consistency degree, and utility value weights through a numeric exercise. Section 4 presents the results and comparative analysis. Finally, Section 5 provides conclusions.

## 2. Current Issues

The quality-adjusted life years (QALY) is calculated as the expected utility value multiplied by life duration in particular health status. $QALY=u_{i}\times \left(Years\right)$ has emerged as a standard outcome indicator to find the best alternatives in health economic assessment, and the QALY’s cost is used as the metric for evaluating cost-effectiveness thresholds. Conventionally, the time trade-off simple (TTO) and composite (cTTO) techniques have been applied to elicit preferences since 1970 (Torrance) [14,15,16,17,18,19,20,21,22,23,24,25,26,27], following expected utility theory (EUT) in the Von Neumann-Morgenstern tradition [28] with a set of closed and bounded options, which assume an evenly distributed social and economic environment. On the other hand, academic interest has been focused on improving the statistical parametric model’s robustness to obtain utility values, going from generalized least squares (GLS) with random and fixed effects to the Tobit model with censored dependent variables. In other cases, to reduce the biased QALY weights, the prospective theory is used, and experienced utility theory is implemented. However, in these models, only one utility value is assigned to health status, without considering that a single health status code from the EQ-5D questionnaire represents different individuals with dissimilar perceptions of their health status.

The EQ-5D-5L survey describes a health status as perceived in five dimensions (5D): mobility (MO); self-care (SC); usual activities (UA), pain/discomfort (PD), and anxiety/depression (AD), with five response levels (5L), from no problems (1) to an extreme degree of problems (5). These dimensions described the World Health Organization’s (WHO) definition of health in 1946—a state of complete physical, mental, and social well-being, not merely the absence of disease or infirmity [29]—and the response levels describe the declared disability related to the health status. Thus, a 5-digit number describes the perceived health status of an individual. In total, it is possible to generate 3125 theoretical health statuses, from 11111 (the best) to 55555 (the worst). Thus, the WHO’s definition is considered to be the decision problem objective, and the health dimensions to be the criteria that are predefined and clearly stated under a multi-criteria decision-making analysis (MCDM) paradigm. Additionally, the standardized valuation study protocol (called EQ-VT) is used to obtain utility values $\left(u_{i}\right),$ and its structure allows the measurement of each alternative’s performance and the calculation of the criteria weight.

The EQ-5D questionnaire includes a vertical visual analog scale (VAS), which measures the declared health status from the worst (0) to best (100). This quantitative measurement of a health outcome reflects an individual judgment at a specific time, capturing the subjective experience in the health-illness process, showing variability in preferences and, therefore, allowing for the association of utility levels related to individual characteristics [30]. Therefore, the traditional QALY calculation is used.

Although alternative MCDM methods have been tested to derive utility values, and they have been used in hospital management, theoretical descriptions, and the analysis of the course of action for specific diseases, such as cancer [31,32,33,34], they have not been applied to the analysis and derivation of utility values obtained through the use of the EQ-5D-5L.

Among MCDM techniques, there are several approaches with different levels of complexity and theoretical bases to elicit preferences and utility values. In this sense, taking into account the decision process context in this study and the type of results the chosen method is expected to bring—that is, the numerical value (utility or score)—methods, such as MAUT [35], the simple multi-attribute rating technique (SMART) [36], AHP [37,38], the measuring attractiveness by a categorical-based evaluation technique (MACBETH) [39], and technique for order preference by similarity to the ideal solution (TOPSIS) [40] could be used. However, while MAUT uses a compensation process (e.g., between the quality and quantity of life), the other techniques estimate weights more simply, and their primary data collection is not as expensive; for this reason, these are recommended for use in low- and low-middle-income countries [41,42,43].

Németh et al. 2019 [44] compared several weighting methods used in MCDM in health care: direct weighting, AHP, conjoint analysis (CA), discrete choice experiments (DCE), MACBETH, potentially all pairwise rankings of all possible alternatives (PAPRIKA), and SMART. This comparison was made using resource requirements, software requirements, the chance of bias, and the general complexity, and the conclusion was that AHP was an appropriate method to use, considering that it had a moderate resource requirement and the lowest level of complexity and might be more suitable to explore preference elicitation methods based on weight dimensions. This additive method converts subjective assessments of relative importance to a vector of priorities and is based on pairwise comparisons performed within each comparative criterion; judgments are made using the Saaty scale, and consistency is applied to check the transitivity axiom based on the decision maker´s judgments. In our study and following authors [45,46,47], we assumed that medical decision-making needs to consider trade-offs between health dimensions, and the AHP provides a framework that can help decision-makers understand the trade-offs made between dimensions in individual health status perception.

## 3. Materials and Methods

### 3.1. Study Design

This observational, descriptive, and non-experimental study used a non-probabilistic quota sampling from the community of the National University of Colombia at Medellin (Universidad Nacional de Colombia—Sede Medellín). The quotas were segmented into mutually exclusive subgroups of students, professors, and administrative staff based on a specified proportion of the population. In each quota, every individual had an equal and nonzero chance of being included in the study, and the selection of one person in a quota did not affect the inclusion or exclusion of other persons. The sampling selection within each quota was random to minimize selection bias. Hence, the comparison groups differed in their perception and preferences for different health states.

The final sampling reflected the heterogeneity of the population in terms of age, socioeconomic status, and educational level. The inclusion criteria considered active members of the university community who were over 18 years old and agreed to participate in the study. The sample included 301 adults with ages ranging between 18 and 90 years old. The fieldwork period was between June and December of 2017, the university healthcare service and the Universidad Nacional de Colombia approved the ethical component of the study under cods M-USS-0236 and CEMED-078-19 respectively. 

The research constituted a methodological test, and it was not possible to infer causality or perform an association analysis from it; however, the methodology could be applied to any population or country and included useful measures for qualitative health status assessment, such as patient preferences with respect to the healthcare process and delivery of health services. Additionally, it provided rigorous research strategies that allowed going from individual assessments to the population, without going through an aggregate weighted sum that conceals behavior at the individual level; it could be used with generic surveys at the international level or adapted to specific questionnaires in terms of interventions, territorial contexts, or health services.

The survey, applied through personal interviews, included quantitative and qualitative questions in four blocks. In the first block, participants were asked about social and demographic characteristics. In the second block, health dimensions were measured following the printed version of the EQ-5D-5L, authorized by EuroQol under the 27819 code. The third block included the VAS, where the individuals stated their subjective perception of their health status. This information was later used as a reference value to evaluate the proposed hybrid method. Finally, each person created a pairwise comparison matrix with $\frac{n\left(n-1\right)}{2}$ judgments [48]. Such judgments incorporated intangible and tangible factors in the declared preferences, and the weights of each dimension at the individual and aggregate levels were calculated. The dimensions weights were used to form a health index or scale, and their internal consistency was tested through Cronbach’s alpha coefficient for each method, following $\alpha =\frac{n}{n-1}\left[1-\frac{\sum^{\;}v_{i}^{2}}{v^{2}}\right]$, where n is the number of the health dimension, $v_{i}^{2}$ is the variance of the ith item, and $v^{2}$ is the variance of the total score formed by summing all dimensions. [49,50].

### 3.2. The Proposed Model

This study proposed a hybrid methodology using the simple AHP, AHP with a D-number extended fuzzy preference relation, and fuzzy AHP. Through a pairwise comparison matrix, each person revealed the relative importance of one dimension versus the others in terms of the perception of his/her health status and assigned a relative importance value following the simple Saaty scale; then, the weights and utility values were generated for each dimension through the eigenvector process. Afterward, each 301 eigenvector, obtained by the AHP approximation, was combined with the five-digit code that described each health status through the technique for order performance by similarity to the ideal solution (TOPSIS). Thus, the utility values, between zero and one, were calculated as the Euclidean distances from the health declared status of each person to the ideal (11111) and the anti-ideal (55555). This process included the disability associated with each dimension that was captured through the response levels of the EQ-5D-5L survey dimensions (see Table 1).

A schematic diagram of the proposed hybrid model is presented in Figure 1. Under the proposed methodology, the validity of the results using a predefined set of criteria aimed to enable decision-makers to solve conflicting real-world quantitative and/or qualitative multi-criteria problems and to find best-fit alternatives from a set of alternatives in specific, uncertain, fuzzy, and risky environments. In this sense, to show the reliability and robustness of the outcomes in this study, the chosen methods allowed us to obtain the same result so that they could be compared, meaning that the numerical values (utility and score) were compared through the consistency test, aggregate weights, hierarchy of dimensions, and Euclidean distance criterion [51,52].

#### 3.2.1. Analytic Hierarchy Process (AHP)

There were several approaches with different levels of complexity and theoretical bases to elicit preferences from stakeholders as patients. Since health dimensions often overlap conceptually, the analytic hierarchy process (AHP) might be more suitable for exploration in preference elicitation methods to generate preference-based weights dimensions. It is an additive method that converts subjective assessments of relative importance to a vector of priorities, based on pairwise comparisons, where, to build a matrix of order *n*, each person defined the relative importance for each dimension in the assessment of their health status at the time of the survey through 10 judgments adding to the EQ-5D-5L survey; then, build a 5 × 5 comparison matrix individual and aggregate reciprocal with all leading elements unity in the main diagonal. These judgments were made using the Saaty scale, and the consistency principle was applied to check the transitivity axiom on the decision maker’s judgments. This method was useful for studying complex decision-making problems and one of the most popular amongst the MCDM; it was used to calculate the value of individual judgments in the decision-making process so that they might be aggregated. Given n dimensions and k decision-makers, a typical multi-criteria decision-making problem was expressed in matrix format as:(1)$$A=\begin{array}{c} MO\\ SC\\ UA\\ \begin{array}{l} PD\\ AD \end{array} \end{array}\left[\begin{array}{ccccc} MO & SC & UA & PD & AD\\ 1 & a_{12} & a_{13} & a_{14} & a_{15}\\ a_{21} & 1 & a_{23} & a_{24} & a_{25}\\ a_{31} & a_{32} & 1 & a_{34} & a_{35}\\ a_{41} & a_{42} & a_{43} & 1 & a_{45}\\ a_{51} & a_{52} & a_{53} & a_{54} & 1 \end{array}\right]$$
where $D_{i}$ denotes each health status dimension. The positive numerical value $a_{ij}$ measured the relative importance of each dimension in their perceived health status. It took the numerical values on the Saaty scale $a_{ij}=x$ and (1≤x≤9) (see Table 1). Priorities were calculated using the eigenvector method. Reciprocity was tested when $a_{ij}=x$ and $a_{ij}=1/x$
∀ i≠j, $a_{ij}=1$
∀ i=j. In total, each individual made (n (n−1))/2 judgments to build the matrix |A| with n = 5. The weights and scores, called priorities, were derived from the pairwise comparison matrix |A|, where k decision maker compares two health dimensions at the same level of the hierarchy.

After the normalization of |A|, we calculated the priority weights using the geometric mean and assigned the same weight to all individuals in the group [53] $Aw=\lambda_{max}w$, where $\lambda_{max}$ is the eigenvalue, and (ω) the eigenvector $\omega ={\left(\omega_{1},\;\omega_{2},\omega_{3},\omega_{4},\omega_{5}\right)}^{T}$ of |A|. Finally, we tested the axiom of transitivity of preferences using the consistency ratio (CR) (Equation (2)). If CR<0.10, the consistency degree was acceptable, and the eigenvector could be used as the weighting vector. $CR=\frac{CI}{RC}$, where $CI=\left(\lambda_{max}-n\right)/\left(n-1\right)$, and RC is the random consistency associated with the matrix size. In this case, for n=5, the AC=1.12 (see Table 1).

#### 3.2.2. AHP with a D-Number Extended Fuzzy Preference Relation (D-AHP)

The theory of evidence, considered a generalization of the Bayesian theory of subjective probability, is a structure for modeling uncertainty and perceptions through a belief function based on rules, known as the Dempster–Shafer theory (D-S) [54,55]; it has two fundamental goals: i) obtaining credibility values from subjective probabilities and ii) modeling random and epistemic uncertainty [56]. In 2012, Deng [57] proposed D-number theory as an extension of the D-S theory, incorporated no excludable hypothesis, enabling the modeling of complete and incomplete information settings in the framework of discernment and taking into consideration that human valuations inevitably have intersections. In the mathematical framework of D-S, the basic probability assignment (BPA) expressed uncertainty in judgment by assigning probabilities to a subset composed of N objects rather than to an individual object. In 2014, Deng [58] combined the AHP with D-numbers according to the following description: the transform matrix |A| in the fuzzy preference relation [59] follows Equation (2).
(2)$$r_{ij}=g\left(a_{ij}\right)=\frac{1}{2}\left(1+log_{9}a_{ij}\right)$$

Thus, the pairwise comparison matrix was transformed, considering linguistic values, into a fuzzy preference relationship called R matrix, where the dimensions $D=\left\{D_{1},D_{2},\ldots ,D_{n}\right\}$ are represented by a fuzzy set in DxD, characterized by a membership function ${\tilde{\mu }}_{R}:D\;x\;D\to \left[0,1\right]$. In this exercise, the D matrix had complete information, and their cardinality was small, and so, the preference ratio could be represented by the $R={\left|r_{ij}\right|}_{nxn}$ matrix, where $r_{ij}=\mu_{R}\left(a_{i},a_{j}\right)\;\forall_{i,j}\in \left\{1,\;\ldots ,n\right\}$, and the probability rules could be applied to construct the Rp matrix. The degree of preference of the dimension $D_{i}$ over *D_j_* was: $r_{ij}=\frac{1}{2}$ indicated that they were indifferent, and the associated probability was *Pr* = 0; if $r_{ij}<\frac{1}{2},$ then *D_j_* was preferred to $D_{i},$ and Pr=0. If $r_{ij}=1$, then $D_{i}$ was absolutely preferred to $D_{j},$ and *Pr* = 1; if $r_{ij}>\frac{1}{2}$, *D_i_* was preferred to *D_j_*, and Pr=1, Preference probabilities by dimensions were evaluated to construct the probability matrix of ones and zeros. Thus, the dimensions were sorted using the triangulation method, as follows:Calculated the sum by rows.Deleted the row and column with the highest number derived in step i. The first to be removed represented the most preferred dimension, and so on.Repeated (i) and (ii) until the probability matrix was empty. Constructed a new *R* matrix, considering the order of the eliminated rows called $R_{c}.$

The dimension’s weights $\left(w_{i}\right)$ were calculated solving the equation system at the upper triangular matrix in $R_{c}$ under the following restrictions: $\sum_{i=1}^{n}w_{i}=1$; $w_{i}\geq 0$ and λ>0. λ represented the expert’s cognitive capability or credibility levels in regard to declared preferences at the matrix A. Thus, the system of equations was solved for different levels of λ; the lowest value represented the highest credibility level $\left(\underline{\lambda }\right)$, (λ=n) represented a medium credibility level, and lower credibility was represented by $\left(\lambda =n^{2}/2\right)$.

#### 3.2.3. Fuzzy AHP (F-AHP)

Decision-making in a fuzzy environment is understood as a process with both restrictions and consequences that are not known with certainty and whose limits are not clear [60]; in contrast to the concept of probability; however, humans can process this fuzzy information, as well as follow fuzzy instructions [61]. In this study, the decision-maker expressed their judgment on their health status and the relative importance of this perception in linguistic terms, making them vague, ambiguous, and subjective [62]. Thus, the Saaty scale was extended to fuzzy numbers, creating Fuzzy AHP. A fuzzy trapezoidal number $\tilde{A}=\left(a,\;b,\;c,d\right)$ was represented as the membership function $\mu_{\tilde{A}}\;\left(x\right)$, given by Equation (3).(3)
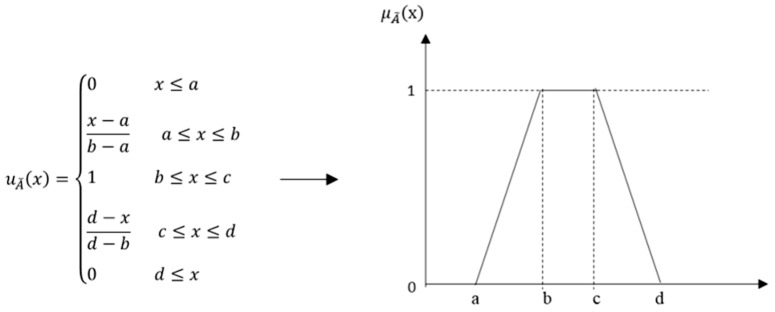


The dimension weights were aggregated using the Yager method (1981) [63], and the crisp numbers were calculated using α cuts. This way, after transforming the pairwise comparison matrix, priorities could be calculated for each α cut, creating priority vectors using a group of crisp comparisons or intervals and maximizing the satisfaction of the decision-makers through a specific crisp priority vector. The expert’s judgment in terms of the relative importance between dimensions i and j was calculated via a geometric mean using Equation (4).
(4)$${\tilde{r}}_{ij}={\left({\tilde{a}}_{ij1}{\tilde{a}}_{ij2}\cdots {\tilde{a}}_{ijk}\right)}^{\frac{1}{k}}$$
where ${\tilde{a}}_{ijk}$ denotes the paired comparison between health dimensions $D_{i}$ and $D_{j}$ for the person/expert k. Afterward, the aggregated matrix of fuzzy numbers ${\tilde{r}}_{ij}$ was transformed into a matrix of crisp numbers $r_{ij}$ using Equation (5).
(5)$$r_{ij}=\int_{0}^{1}\;\frac{1}{2}\left({\left({\tilde{r}}_{ij}\right)}_{\alpha }^{L}+{\left({\tilde{r}}_{ij}\right)}_{\alpha }^{U}\right)d\alpha$$

If the trapezoidal fuzzy number ${\tilde{r}}_{ij}$ was parameterized as (a,b,c,d), then:(6)$${\left({\tilde{r}}_{ij}\right)}_{\alpha }^{L}=a+\left(b-a\right)\alpha$$
(7)$${\left({\tilde{r}}_{ij}\right)}_{\alpha }^{U}=d+\left(d-c\right)\alpha$$

Replacing (6) and (7) in Equation (5), the crisp number for each position of the paired comparison matrix was:(8)$$r_{ij}=\int_{0}^{1}\;\frac{1}{2}\left[a+\left(b-a\right)\alpha +d+\left(d-c\right)\alpha \right]\;d\alpha$$

To calculate individual values and following Zheng [64], two trapezoidal fuzzy numbers ${\tilde{A}}_{1}$ and ${\tilde{A}}_{2}$ parameterized as $\left(a_{1},b_{1},c_{1},d_{1}\right)$ and $\left(a_{2},b_{2},c_{2},d_{2}\right),$ respectively, the next properties should be applied in order to calculate the fuzzy weights ${\tilde{w}}_{i}.$
(9)$${\tilde{A}}_{1}⊕{\tilde{A}}_{2}=\left(a_{1},b_{1},c_{1},d_{1}\right)⊕\left(a_{2},b_{2},c_{2},d_{2}\right)=\left(a_{1}+a_{2},b_{1}+b_{2},c_{1}+c_{2},d_{1}+d_{2}\right)$$
(10)$${\tilde{A}}_{1}⊗{\tilde{A}}_{2}=\left(a_{1},b_{1},c_{1},d_{1}\right)⊗\left(a_{2},b_{2},c_{2},d_{2}\right)=\left(a_{1}a_{2},b_{1}b_{2},c_{1}c_{2},d_{1}d_{2}\right)$$
(11)$${\left({\tilde{A}}_{1}\right)}^{-1}=\left(\frac{1}{a_{1}},\frac{1}{b_{1}},\frac{1}{c_{1}},\frac{1}{d_{1}}\right)$$

Once the fuzzy geometric mean value by health dimensions ${\tilde{r}}_{i}$ was calculated, and the weights ${\tilde{w}}_{i}={\tilde{r}}_{1}⊗{\left({\tilde{r}}_{1}⊕{\tilde{r}}_{2}⊕{\tilde{r}}_{2}⊕{\tilde{r}}_{4}\right)}^{-1}$ were determined, then the geometric mean was multiplied by weights in order to convert the trapezoidal fuzzy number into crisp values (N), using the Center of Area Value (CAV) as: (12)$$N=\frac{b+c}{2}+\frac{\left[d-c-\left(b-a\right)\right]}{2}=\frac{a+2b+2c+d}{6}$$

All matrices took a simple or transformed Saaty scale (see Table 1). Thus from AHP, D-AHP to F-AHP, it was possible to find individual and aggregate priority vectors.

#### 3.2.4. The Technique for Order Performance by Similarity to the Ideal Solution (TOPSIS)

This method obtained the utility values of each health status code on a 0-to-1 scale. Although there was no choice of alternatives, the dimension qualification defined the health status of each person. Thus, the rows of the pairwise comparison matrix (301 individuals and 5 dimensions) represented the interviewed subject (E) and the columns containing the value assigned by the subject to each health dimension; so, $x_{ij}$ represents the personal assessment $E_{k}$ regarding dimension $D_{j},$ with k=1, 2… 301 and j=1,2… 5. In this sense, $x_{53}=2$ meant that person 5 believed that she had slight problems carrying out her daily activities. To normalize the matrix of criteria $D^{k}$, each dimension qualification was calculated for each dimension, that is, the total value per column (Equation (13)).
(13)$$r_{ij}=\frac{x_{ij}}{\sqrt{\sum_{j=1}^{301}\;{\left(x_{ij}\right)}^{2}}}$$

To calculate the normalized weights, each criterion $r_{ij}$ was multiplied by the aggregated and individual priority vector as: $v_{ij}=r_{ij}\omega$. In this case, 𝜔 represents the priority vector, calculated using three different methods: AHP, D-AHP, and F-AHP. Thus, three different health indices or utility values for the declared health status were calculated. In the case of D-AHP, we took the weights vector corresponding to a high credibility level $\left|\underline{\lambda }\right|$.

Now, given that the health code 11111 represents the best possible health status, we assumed it as the ideal value $\left(v_{j}^{+}\right),$ and the 55555 as the anti-ideal value $\left(v_{j}^{-}\right)$. With these values, following Equation (14), we calculated for a person $E_{i}$, the Euclidean distance from the declared value regarding the ideal and anti-ideal health status.
(14)$$S_{i}^{+}={\left[\sum_{j=1}^{5}\;\left(v_{ij}-v_{j}^{+}\right)\right]}^{\frac{1}{2}}\;\mathrm{and}\;S_{i}^{-}={\left[\sum_{j=1}^{5}\;\left(v_{ij}-v_{j}^{-}\right)\right]}^{\frac{1}{2}}$$

With this information, the performance weight or value associated with each health status code for each one of the 301 respondents was estimated as:(15)$$p_{i}=\frac{S_{i}^{-}}{S_{i}^{+}+S_{i}^{-}}$$

The value between 0 and 1 associated with each declared health status and represented by a 5-digit code allowed us to classify them from best to worst, in descending order for the *p_i_* value [65]. 

#### 3.2.5. Numerical Example

This numerical example is presented to illustrate the proposed method and find health status utility values. The decision problem located the health status dimensions at the criteria level, allowing the identification of some aspects of the decision problem. With this information, several alternatives could be evaluated using those criteria values, and the preference relation, which was established to express the experts’ judgment at the criteria level, was a crucial point. The D-AHP process is illustrated with information to the person as (Figure 2):

To obtain weights through F-AHP, only one comparison matrix is illustrated; the process followed the same steps with the 301 matrices (Figure 3). 

The aggregate F-AHP example took information from five people and illustrated how to calculate the aggregate crisp number, taking the paired comparison between mobility (MO) and self-care (SC) as follows (Figure 4):

The process of combining AHP, D-AHP, and F-AHP with TOPSIS is illustrated by taking the AHP matrix. 

The results of the AHP, D-AHP, and F-AHP combined with TOPSIS were compared with VAS declared values in terms of the correlation coefficient in order to determine which one generated the best fit for individual perception. Thus, the Euclidean distance criterion (ED) (Equation (16)) could test which method provided the best fit according to perfect consistency; this was calculated for all components of the aggregated matrix and their prioritization vector, and the minimum ED value allowed us to select the best model [66].
(16)$$ED={\left[\underset{i}{\sum }\;\underset{j}{\sum }\;{\left(a_{ij}-\frac{\omega_{i}}{\omega_{j}}\right)}^{2}\right]}^{\frac{1}{2}}$$

## 4. Results and Discussion

According to the examined declared health status, the pain/discomfort and anxiety/depression dimensions showed the highest disability levels (see Table 2) and the 80% of the statuses were concentrated in fourteen codes (Table 3); perfect health (11111) was the most frequent (34%). The sample included adults between 18 and 90 years old, mostly men (56.15%), with complete or incomplete university studies; 40% earned an income above three times the Colombian legal minimum wage in 2017 (USD 255,4), 58% were from the middle and lower-middle class, and more than 60% did not participate in programs promoted by their healthcare insurance. In the mobility dimension, statistically significant differences were found by χ^2^ in age, sex, educational level, and social-economic status (p<0.05); in pain/discomfort by age (p≤0.01); in distress/depression by age (p≤0.001); in educational level (p≤0.001); and in income level (p≤0.001). Finally, 79% of the interviewees associated the self-care and usual activities dimensions with the capacity of mobility; therefore, no significant differences were found in demographic characteristics, which could show the possibility of dependence among the health status dimensions. 

Related to the consistency of preferences for the 301 AHP matrices, RC<0.10 was found in 283 matrix preferences, 16 had the perfect consistency, and two had small disturbances regarding the perfect consistency of $\lambda_{max}-n$, equal to 0.119 and 0.139, so their values were considered appropriate. In D-AHP, the mean and median of the high credibility levels were 2.6; the standard deviation was 1.33, the minimum was 0.16, and the maximum was 5 for 2.3% of the individuals. High credibility levels above the mean were typical in individuals between the ages of 38 and 50 with an educational level lower than college. Below the mean were people over 50 years old with a college or higher-level education; these differences were statistically significant at P=0.05 and P≤0.001, respectively.

The subjective utility values by health status (see Table 3) showed the aggregate pairwise comparison matrices by hybrid methods, the aggregate weight values, the ideal reference point, and the anti-ideal scenarios. Mobility had the shortest distance from the ideal point and the longest from the anti-ideal reference point, followed by pain/discomfort. In other words, a utility value closer to 100 in the health status was associated with no pain/discomfort and no mobility problems. The hierarchy and weights of the other dimensions depended on the method used. For AHP and F-AHP, the ranking of the dimensions was self-care, usual activities, and anxiety/depression. In all methods, the ordinal preferences preserved rank strongly, which implied that the elements of the pairwise comparison matrix exhibited $a_{ij}\geq 1$ and $\omega_{i}\geq \omega_{j}$. Hence, we could prove that this method for health status assessment was a useful technique to measure preference consistency.

Euclidean distance (ED) was calculated following Equation (16), considering the matrix components added by each method, which resulted in AHP-TOPSIS = 0.54, D-AHP-TOPSIS = 0.57, and F-AHP-TOPSIS = 0.50, the latter being the value that represented the best approximation to ordinal preferences. The Cronbach’s alpha coefficients were 0.74, 0.78, and 0.81 for D-AHP-TOPSIS, AHP-TOPSIS, and F-AHP-TOPSIS, respectively, and we could conclude that the questionnaire had satisfactory internal validity.

Finally, since each person had an associated value for the credibility level and dimension weights according to their declared health status, it was possible to find a value corresponding to that status through the stated preferences in the pairwise comparison matrix rather than only one utility value for a five-digit code, independent of the individual preference perception, as happens in the traditional estimation. In Table 4, we presented the results of the most frequent codes in terms of the mean, standard deviation, minimum, and maximum values, as well as the worst declared health status. 

The graphs (Figure 5) compared the individual utility values with the real perceptions declared by VAS using each of the proposed methods. The 45° line showed the VAS results and the distance between the declared and estimated utility values, which allowed us to establish the best method. In this way, D-AHP-TOPSIS proved to be the method that presented the best fit when the health status was closer to one (1) or zero (0), and the individual values from D-AHP-TOPSIS presented a better fit to the mean value according to the VAS. However, by employing the paired correlation coefficient at a 5% level of significance, the relation degree of these quantitative and continuous variables was found to be higher in F-AHP-TOPSIS = 0.697, D-AHP-TOPSIS = 0.543, and AHP-TOPSIS = 0.648. Finally, the best adjustment of the aggregate preferences of the hybrid methods to the perception by VAS for all health statuses was better in F-AHP-TOPSIS.

## 5. Conclusions

The hybrid method approach has some attractive properties, such as a low-cost survey process, a simple mathematical algorithm, a natural consistency index, rank preservation, and precision, and it can be used as an alternative to the known prioritization methods. Additionally, it can be applied in certain medical decisions concerned with understanding how a specific person values his or her current health or possible health outcomes from different interventions in small population samples, and it is particularly useful for studies carried out in low- and middle-low-income countries, which do not have a population value set for the QALY estimation and use information from another country.

In this research, we proved that it was possible to calculate individual and aggregate assessments of utility values and not only a value adjusted to the sample average. The computational algorithm was simple and did not require specialized software or long processes and training costs for interviewers. It helped to eliminate bias due to the time duration in different health statuses evaluated through different scenarios, as it occurred with the cTTO, and it could be carried out in different cultural and health system contexts to classify a health status in terms of preferences and dimensions of specific populations with different health programs. Additionally, it was possible to calculate the variability of the health status declared by several people under the same code, enabling the extension of the analysis to 3125 theoretical states. 

The hybrid methodology provided several contributions: (i) Through individual preferences, it was possible to assign a utility value for each declared health status. (ii) Considering the response level given to each dimension in TOPSIS provided the current level of disability; thus, this point reflected the individual characteristics of the person performing the valuation and became the essential reference point for obtaining utility values, and the judgments reflected the health and illness experience over time. (iii) The results provided useful information, especially for low- and middle-low-income countries, where it is recommended to use simple methods that are not as expensive in primary data collection. 

When studying the response to EQ-5D, considering it as a linguistic variable, a better fit for the consistency among preferences and the quantitative value associated with the health status declared through VAS was obtained. However, these findings need to be validated in future research by exploring the incorporation of uncertainty and inaccuracy in the decision-making process by individuals with different initial health statuses. Thus, this analysis must begin by taking into account individual perceptions and then aggregating them by common characteristics, which aims to indicate equity issues in low- and middle-low-income countries. Furthermore, including the regional context could reduce the cognitive bias related to the stated preferences affected by the local health system [67].

In this study, a choice between alternatives did not take place; only the level of criteria assessment (AHP) was taken as a necessary input for calculation in the QALY approach. Thus, combining the paired comparison matrix for a specific person with its health code, which includes each dimension’s levels, must identify the current state of disability. This combination allowed us to identify how far the declared health status was from the ideal health status, represented by code 55555 (TOPSIS). Under the proposed methodology, the validity of the results using a predefined set of criteria aimed to enable decision-makers to solve conflicting real-world quantitative and/or qualitative multi-criteria problems and to find best-fit alternatives from a set of alternatives in specific, uncertain, fuzzy, or risky environments. Although for the current research, the AHP method was selected as the baseline weighting method, the authors recognized that there were other methods, such as Paprika and Ca, that were more complex and had higher resource requirements that could be used to refine the overall methodology; this could be a topic for future work [68].

## Figures and Tables

**Figure 1 ijerph-17-01423-f001:**
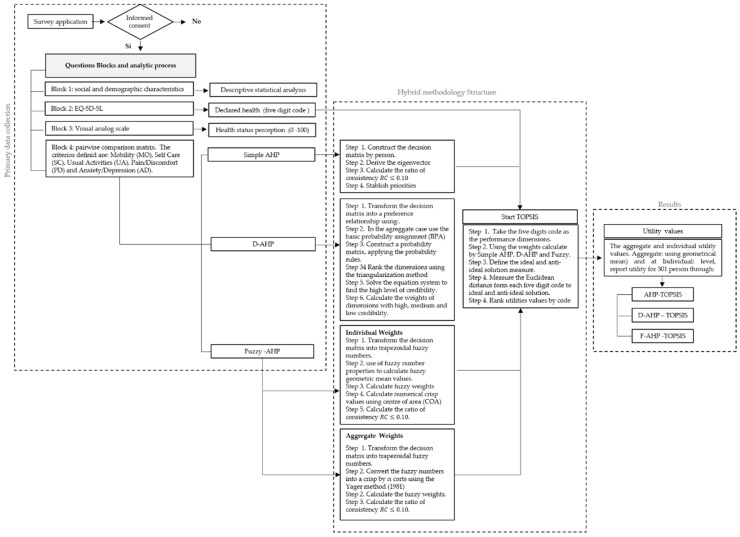
Data collection and schematic diagram of the proposed hybrid model.

**Figure 2 ijerph-17-01423-f002:**
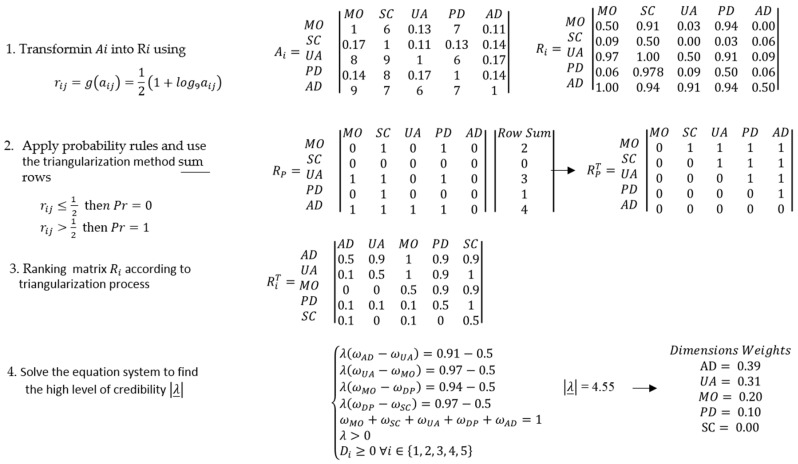
D-AHP numerical example process.

**Figure 3 ijerph-17-01423-f003:**
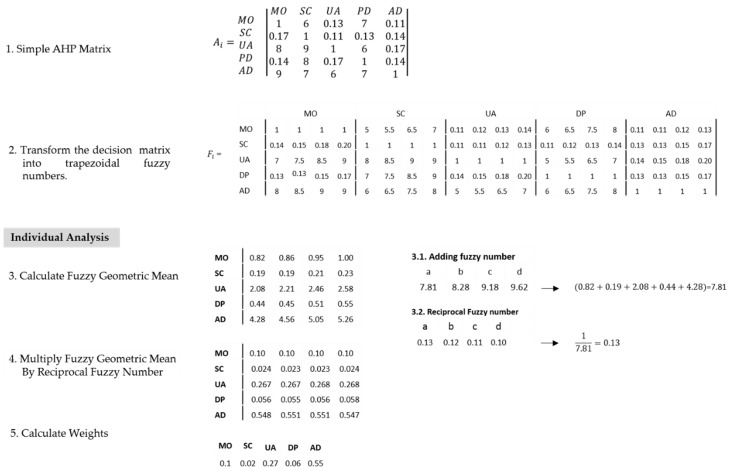
F-AHP numerical example process to individual analysis.

**Figure 4 ijerph-17-01423-f004:**
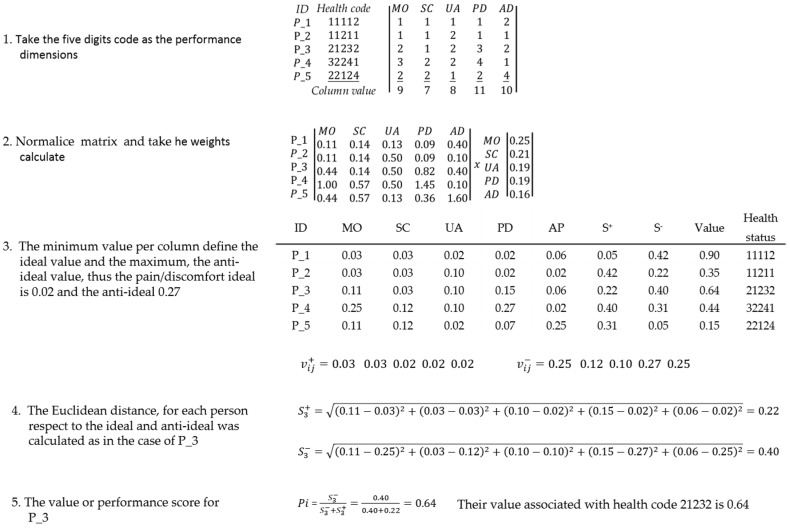
F-AHP numerical example process to aggregate analysis.

**Figure 5 ijerph-17-01423-f005:**
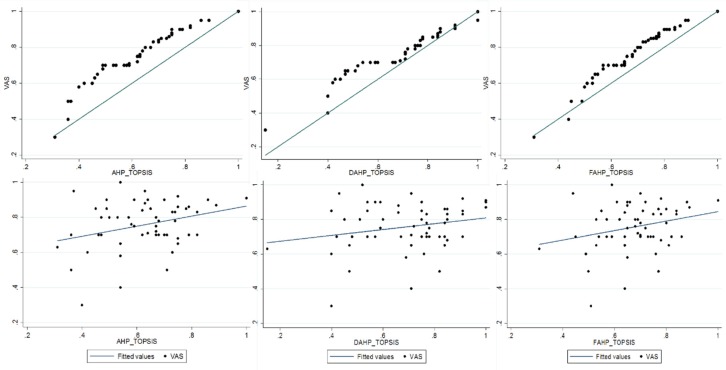
Adjustment hybrid methods to perception by visual analog scale (VAS).

**Table 1 ijerph-17-01423-t001:** Numerical rating in the analytic hierarchy process (AHP) and trapezoidal fuzzy numbers associated.

Scale	Numerical Rating		Reciprocal (Decimal)		Fuzzy Trapezoidal			Fuzzy Reciprocal (Inverse)		
Equal importance	1		1 (1000)		(1, 1, 1, 1)			(1, 1, 1, 1)		
Moderate importance	3		1/3 (0.33)		(2, 5/2, 7/2, 4)			(1/4, 2/7, 2/5, 1/2)		
Strong importance	5		1/5 (0.20)		(4, 9/2,11/2,6)			(1/6, 2/11, 2/9,1/4)		
Very strong importance	7		1/7 (0.14)		(6, 18/2, 15/2, 8)			(1/8, 2/15, 2/13, 1/6)		
Extreme importance	9		1/9 (0.11)		(8,17/2, 9, 9)			(1/9, 1/9, 2/17, 1/8)		
Intermediate values between two adjacent judgments	2		1/2 (0.50)		$\left(x-1,\;x-\frac{1}{2},x+\frac{1}{2},x+1\right)$			$\frac{1}{\left(x+1\right)},\frac{1}{\left(x+\frac{1}{2}\right)},\;\frac{1}{\left(x-\frac{1}{2}\right)},\frac{1}{\left(x-1\right)}$		
	4		1/4 (0.25)							
	6		1/6 (0.17)							
	8		1/8 (0.13)							
Size matrix	1	2	3	4	5	6	7	8	9	10
Random consistency (RC)	0	0	0.58	0.9	1.12	1.24	1.32	1.41	1.45	1.49

Adapted [39,58].

**Table 2 ijerph-17-01423-t002:** Background characteristics of the sample.

Characteristics		*n* (%)
Sex	Female	132 (43.5)
	Male	169 (56.15)
Age	18–21	76 (25.25)
	22–47	77 (25.58)
	48–66	79 (26.25)
	67–90	69 (22.92)
Education levels	Primary	29 (9.63)
	Secondary	38 (12.62)
	Bachelor degree	50 (16.61)
	Bachelor student	105 (34.88)
	Professional technician	11 (3.65)
	Technologist	17 (5.65)
	Master degree	39 (12.96)
	Ph.D.	12 (3.99)
Wage	<1 minimum wage	78 (25.91)
	1 < minimum wage < 2	104 (34.55)
	>3 minimum wage	119 (39.53)
Social-economic status	1 (lowest)	7 (2.33)
	2	51 (16.94)
	3	113 (37.54)
	4	63 (20.93)
	5	51 (16.94)
	6 (highest)	16 (5.32)
Health program	Yes	93 (30.90)
	No	206 (68.44)
Mobility	No problems	246 (81.73)
	Slight problems	32 (10.63)
	Moderate problems	18 (5.98)
	Severe problems	5 (1.66)
Self-care	No problems	298 (99)
	Slight problems	3 (1)
Usual activities	No problems	253 (84.05)
	Slight problems	39 (12.96)
	Moderate problems	8 (2.66)
	Severe problems	1 (0.33)
Pain/discomfort	No pain or discomfort	172 (57.14)
	Slight pain or discomfort	78 (25.91)
	Moderate pain or discomfort	45 (14.95)
	Severe pain or discomfort	5 (1.66)
	Extreme pain or discomfort	1 (0.33)
Anxiety/depression	No anxiety or depression	183 (60.80)
	Slight anxiety or depression	74 (24.58)
	Moderately anxiety or depression	38 (12.62)
	Severely anxiety or depression	4 (1.33)
	Extremely anxiety or depression	2 (0.66)

**Table 3 ijerph-17-01423-t003:** Aggregated pairwise comparison matrices and dimensions weights.

Models	Dimensions						TOPSIS	
**AHP**	**MO**	**SC**	**UA**	**PD**	**AP**	**Weights (ω)** **^a^**	**Ideal (S^+^)**	**Anti-ideal (S^−^)**
**MO**	1	1.5	1.2	1.2	1.6	0.25	0.01	0.04
**SC**	0.7	1	1.2	0.8	1.1	0.19	0.01	0.03
**UA**	0.8	0.9	1	1	1.3	0.19	0.01	0.03
**PD**	0.8	1.2	1	1	1.3	0.21	0.01	0.02
**AP**	0.6	0.9	0.8	0.8	1	0.16	0.01	0.03
**D-AHP**	**MO**	**SC**	**AU**	**PD**	**AP**	(ω) **^b^**	**(S^+^)**	**(S^−^)**
**MO**	0.5	0.6	0.6	0.6	0.6	0.39	0.01	0.06
**SC**	0.5	0.5	0.5	0.5	0.6	0.27	0.01	0.03
**UA**	0.4	0.5	0.5	0.5	0.5	0.22	0	0.04
**PD**	0.5	0.5	0.5	0.5	0.5	0.12	0.01	0.03
**AP**	0.4	0.4	0.4	0.4	0.5	0	0	0
**F-AHP**	**MO**	**SC**	**UA**	**PD**	**AD**	(ω) **^c^**	**(S^+^)**	**(S^−^)**
**MO**	1	1.4	1.1	1	1.3	0.23	0.01	0.04
**SC**	0.8	1	1.1	0.9	1	0.19	0.01	0.03
**UA**	0.9	0.9	1	0.9	1	0.18	0.01	0.03
**PD**	1	1.1	1.2	1	1.2	0.21	0.01	0.02
**AP**	0.8	0.1	0.9	0.9	1	0.18	0.01	0.03

^a^ RC = 0.001. $\lambda_{max}=5.001$. ^b^
$\left|\underline{\lambda }\right|\;$= 0.41. ^c^ RC = 0.01.AHP: Analytic Hierarchy Process; D-AHP: Analytic Hierarchy Process extended by D-numbers; F-AHP: Fuzzy Analytic Hierarchy Process; TOPSIS: Technique for Order Preference by Similarity to the Ideal Solution; MO: Mobility; SC: self-care; UA: usual activities; PD: pain/discomfort; AD: anxiety/depression.

**Table 4 ijerph-17-01423-t004:** Compared individual utility values.

Health Status	*n*	VAS				AHP-TOPSIS				D-AHP-TOPSIS				F-AHP-TOPSIS			
		Mean	Min	Max	SD	Mean	Min	Max	SD	Mean	Min	Max	SD	Mean	Min	Max	SD
11111	102	0.88	0.5	1.00	0.12	0.87	0.82	0.96	0.02	0.89	0.85	0.90	0.01	0.86	0.83	0.91	0.02
11121	35	0.86	0.5	1.00	0.11	0.85	0.72	0.91	0.04	0.85	0.67	0.90	0.05	0.86	0.84	0.91	0.02
11112	33	0.84	0.5	1.00	0.12	0.84	0.71	0.90	0.05	0.86	0.69	0.90	0.04	0.87	0.83	0.91	0.02
11122	15	0.86	0.6	0.95	0.11	0.82	0.76	0.91	0.04	0.84	0.69	0.89	0.05	0.87	0.84	0.91	0.02
11113	12	0.87	0.7	1.00	0.10	0.85	0.77	0.92	0.04	0.86	0.78	0.89	0.03	0.86	0.83	0.90	0.02
11131	9	0.84	0.7	0.90	0.07	0.81	0.68	0.88	0.06	0.80	0.72	0.89	0.05	0.87	0.83	0.90	0.03
21121	9	0.81	0.3	0.95	0.22	0.83	0.76	0.87	0.04	0.83	0.79	0.87	0.03	0.86	0.83	0.89	0.03
11222	5	0.88	0.7	1.00	0.13	0.84	0.78	0.90	0.05	0.84	0.82	0.87	0.02	0.86	0.84	0.88	0.02
11123	4	0.75	0.5	1.00	0.21	0.84	0.80	0.88	0.03	0.85	0.79	0.89	0.04	0.87	0.84	0.89	0.02
11211	4	0.80	0.55	1.00	0.23	0.87	0.81	0.94	0.06	0.87	0.84	0.89	0.02	0.87	0.85	0.88	0.01
11212	4	0.93	0.9	1.00	0.05	0.82	0.74	0.89	0.07	0.84	0.79	0.88	0.04	0.85	0.84	0.86	0.01
21232	4	0.84	0.8	0.90	0.05	0.76	0.71	0.81	0.04	0.76	0.72	0.81	0.05	0.82	0.80	0.84	0.02
31131	4	0.74	0.65	0.80	0.08	0.71	0.62	0.79	0.09	0.67	0.56	0.73	0.08	0.77	0.74	0.90	0.03
11132	3	0.70	0.5	0.90	0.20	0.81	0.71	0.88	0.09	0.81	0.76	0.87	0.05	0.74	0.72	0.76	0.02
42352	1	0.63				0.61				0.62				0.73			

SD: Standard deviation; VAS: visual analog scale.

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
