# Peer review of "Hybrid Methodology to Improve Health Status Utility Values Derivation Using EQ-5D-5L and Advanced Multi-Criteria Techniques"

_ijerph, 2020, doi:10.3390/ijerph17041423_

Round 1

Reviewer 1 Report

I have read the revised manuscript and the authors have revised it as required, generally. Unfortunately, there are still some issues with the revised manuscript.

The abstract is too long and not succinct. The introduction and the review of current issues are still poor written and made the readers that are not familiar with the field feel confused with the significance and necessity of the discussion. The innovations are still not clearly described in the revised manuscript.

Author Response

Dear Reviewer:

We appreciate all your comments and suggestions to improve the paper. The English was editing by the American Journal Expert. We Attached the certificate.

Comment 1:  The abstract is too long and not succinct. The introduction and the review of current issues are still poor written and made the readers that are not familiar with the field feel confused with the significance and necessity of the discussion. The innovations are still not clearly described in the revised manuscript. 

  1. The abstract was modified, and we hope that is more clear and succinct.

Revised Abstract: This paper presents a new approach to the calculation of quality-adjusted life years (QALY) based on multi-criteria decision-making (MCDM) methods and using the EQ-5D-5L questionnaire. The health status utility values are calculated through a hybrid methodology. We combine the analytic hierarchy process (AHP), the AHP with a D-number extended fuzzy preference relation (D-AHP), the fuzzy analytic hierarchy process (F-AHP), and the technique for order preference by similarity to the ideal solution (TOPSIS) to obtain individual and aggregated utility values. The preference data are elicited using a sample of individuals from a Colombian university. In all tested methods, the ordinal preferences were consistent, and the weights were compared using the Euclidean distance criterion (EDC). We identified F-AHP-TOPSIS as the optimal method; its benefits are associated with modeling the response options of the EQ-5D in linguistic terms, it gives the best approximation to the initial preferences according to EDC, and it can be used as an alternative to the known prioritization method. This hybrid methodology is particularly useful in certain medical decisions concerned with understanding how a specific person values his or her current health or possible health outcomes from different interventions in small population samples and studies carried out in low- and middle-low-income countries.

The introduction and the current issues were rewriting, and hope these include all your recommendations and the innovations are clearly describe.

Revised Introduction:

Decision-making is a natural process in which a person chooses one course of action from a finite set, where each action leads to possible states according to a probability distribution associated with their expected utility value [1]. Theoretically, in the healthcare sector, the process does not differ from the way it operates in other fields, but individual characteristics make it difficult to predict how an individual will respond to a specific treatment, program or medication, which adds complexity to the task. Generally, healthcare decision-making is based on economic assessment through a cost-utility or effectiveness analysis. In these studies, utility values are measured through social preferences for health states and are expressed in quality-adjusted life years (QALY). This approach takes expected utility theory and multi-attribute utility theory (MAUT) techniques as the methodological platform for comparing intervention alternatives and allocating resources efficiently [2-10]. The prioritization method to measure utility values uses the EQ-5D questionnaire and applies the time tradeoff (TTO) to elicit preferences and the visual analogue scale (VAS) to establish a reference point. This conventional measurement assigns only a utility value that describes a particular health status and aggregates individual subjective utility values to make objective social decisions. Then, healthcare institutions make decisions based on social preferences obtained from a population sample assuming several theoretical axioms that generate biased QALY weights [11, 12]. However, what is actually needed in making most medical decisions is to understand how a specific person values his or her current health or possible health outcomes from interventions with small samples; the situation worsens when some countries make medical decisions taking utility value sets from other countries’ populations. The present paper explains how advanced in Multi-Criteria Decision Making (MCDM) methods, such as AHP, D-AHP, and F-AHP, offer alternative utility measure methods to assign weights to health status through individual preferences. In these methods, preferences are elicited when each person assesses the relative importance of each health dimension in the EQ-5D survey. These preferences, combined with the disability levels associated with EQ-5D answer levels, reflect individual characteristics and become the essential reference point for the utility value weights, as Bernoulli stated (1954) [13]; also, the judgments show the experience associated with health and illness. Therefore, we propose a hybrid methodology for calculating not only social preferences but also individual preferences. The methods used are easy-to-implement, low-cost primary data collection methods, and it is possible to assign a utility value for each declared health status. This methodology can be applied to small population samples and is very appropriate for low- and middle-low-income countries. The paper structure is as follows: Section 2 states the current issues. Section 3 describes the classical AHP method as well as the D-AHP, F-AHP, and TOPSIS methods; it also describes key concepts and the steps for calculating the criteria priority weights, consistency degree, and utility value weights through a numeric exercise. Section 4 presents the results and comparative analysis. Finally, Section 5 provides conclusions.

Revised Current Issues

The quality-adjusted life years (QALY) is calculated as the expected utility value multiplied by life duration in a particular health status.  has emerged as a standard outcome indicator to find the best alternatives in health economic assessment, and the QALY's cost is used as the metric for evaluating cost-effectiveness thresholds. Conventionally, the time trade-off simple (TTO) and composite (cTTO) techniques have been applied to elicit preferences since 1970 (Torrance) [14-27], following expected utility theory (EUT) in the Von Neumann-Morgenstern tradition [28] with a set of closed and bounded options, which assume an evenly distributed social and economic environment. On the other hand, academic interest has been focused on improving the statistical parametric model's robustness to obtain utility values, going from generalized least squares (GLS) with random and fixed effects to the Tobit model with censored dependent variables. In other cases, to reduce the biased QALY weights, prospective theory is used, and experienced utility theory is implemented. However, in these models, only one utility value is assigned to a health status, without considering that a single health status code from the EQ-5D questionnaire represents different individuals with dissimilar perceptions of their health status.

The EQ-5D-5L survey describes a health status as perceived in five dimensions (5D): mobility (MO); self-care (SC); usual activities (US), pain/discomfort (PD), and anxiety/depression (AD), with five response levels (5L), from no problems (1) to an extreme degree of problems (5). These dimensions described the World Health Organization's (WHO) definition of health in 1947—a state of complete physical, mental, and social well-being, not merely the absence of disease or infirmity [29]—and the response levels describe the declared disability related to the health status. Thus, a 5-digit number describes the perceived health status of an individual. In total, it is possible to generate 3125 theoretical health statuses, from 11111 (the best) to 55555 (the worst). Thus, the WHO's definition is considered to be the decision problem objective and the health dimensions to be the criteria that are predefined and clearly stated under a multi-criteria decision-making analysis (MCDM) paradigm. Additionally, the standardized valuation study protocol EQ-VT is used to obtain utility values  and its structure allows the measurement of each alternative's performance and the calculation of the criteria weight.

The EQ-5D questionnaire includes a vertical visual analog scale (VAS), which measures the declared health status from worst (0) to best (100). This quantitative measurement of a health outcome reflects an individual judgment at a specific time, capturing the subjective experience in the health-illness process, showing variability in preferences and therefore allowing for the association of utility levels related to individual characteristics [30]. Therefore, the traditional QALY calculation is used.

Although alternative MCDM methods have been tested to derive utility values, and they have been used in hospital management, theoretical descriptions, and the analysis of the course of action for specific diseases, such as cancer [31-34], they have not been applied to the analysis and derivation of utility values obtained through the use of the EQ-5D-5L. 

Among MCDM techniques, there are several approaches with different levels of complexity and theoretical bases to elicit preferences and utility values. In this sense, taking into account the decision process context in this study and the type of results the chosen method is expected to bring—that is, the numerical value (utility or score)—methods such as MAUT [35], the simple multi-attribute rating technique (SMART) [36], AHP [37, 38], the measuring attractiveness by a categorical-based evaluation technique (MACBETH) [39], and TOPSIS [40] could be used. However, while MAUT uses a compensation process (e.g., between the quality and quantity of life), the other techniques estimate weights more simply, and their primary data collection is not as expensive; for this reason, these are recommended for use in low- and low-middle-income countries [41-43].

Németh et al. 2019 [44] compare several weighting methods used in MCDM in health care: direct weighting, AHP, conjoint analysis (CA), discrete choice experiments (DCE), MACBETH, potentially all pairwise rankings of all possible alternatives (PAPRIKA) and SMART. This comparison is made using resource requirements, software requirements, the chance of bias, and the general complexity, and the conclusion is that AHP is an appropriate method to use, considering that it has a moderate resource requirement and the lowest level of complexity and might be more suitable to explore preference elicitation methods based on weight dimensions. This additive method converts subjective assessments of relative importance to a vector of priorities and is based on pairwise comparisons performed within each comparative criterion; judgments are made using the Saaty scale, and consistency is applied to check the transitivity axiom based on the decision maker´s judgments. In our study and following authors [45-47], we assume that medical decision-making needs to consider trade-offs between health dimensions, and the AHP provides a framework that can help decision-makers understand the trade-offs made between dimensions in individual health status perception.

Reviewer 2 Report

This is an interesting and rather complex paper, presenting a new approach to evaluate the quality of life, which can be applied to small population samples, being particularly useful for studies carried out in low and middle-low income countries.

I have no major remarks.

English style and grammar need to be improved. There are several errors to be checked and amended.

Abstract. I recommend entering conclusions at the end of the abstract with a clear message for the reader. The same comment applies to the end of Section 5.

Author Response

Dear Reviewer:

We appreciate all your comments and suggestions to improve the paper.

Comment 1: English style and grammar need to be improved. There are several errors to be checked and amended.

The English was editing by the American Journal Expert, we attached the certificate

Comment 2: Abstract. I recommend entering conclusions at the end of the abstract with a clear message for the reader. The same comment applies to the end of Section 5.

The abstract was modified, and we hope that is more clear and succinct. The conclusions was entering at the end of the abstract. The section 5 was modified.

Revised Abstract: This paper presents a new approach to the calculation of quality-adjusted life years (QALY) based on multi-criteria decision-making (MCDM) methods and using the EQ-5D-5L questionnaire. The health status utility values are calculated through a hybrid methodology. We combine the analytic hierarchy process (AHP), the AHP with a D-number extended fuzzy preference relation (D-AHP), the fuzzy analytic hierarchy process (F-AHP), and the technique for order preference by similarity to the ideal solution (TOPSIS) to obtain individual and aggregated utility values. The preference data are elicited using a sample of individuals from a Colombian university. In all tested methods, the ordinal preferences were consistent, and the weights were compared using the Euclidean distance criterion (EDC). We identified F-AHP-TOPSIS as the optimal method; its benefits are associated with modeling the response options of the EQ-5D in linguistic terms, it gives the best approximation to the initial preferences according to EDC, and it can be used as an alternative to the known prioritization method. This hybrid methodology is particularly useful in certain medical decisions concerned with understanding how a specific person values his or her current health or possible health outcomes from different interventions in small population samples and studies carried out in low- and middle-low-income countries.

Revised Conclusions

The hybrid method approach has some attractive properties, such as a low-cost survey process, a simple mathematical algorithm, a natural consistency index, rank preservation, and precision, and it can be used as an alternative to the known prioritization methods. Additionally, it can be applied in certain medical decisions concerned with understanding how a specific person values his or her current health or possible health outcomes from different interventions in small population samples, and it is particularly useful for studies carried out in low- and middle-low-income countries, which do not have a population value set for the QALY estimation and use information from another country.

In this research, we prove that it is possible to calculate individual and aggregate assessments of utility values and not only a value adjusted to the sample average. The computational algorithm is simple and does not require specialized software or long processes and training costs for interviewers. It helps to eliminate bias due to the time duration in different health statuses evaluated through different scenarios, as it occurs with the cTTO, and it can be carried out in different cultural and health system contexts to classify a health status in terms of preferences and dimensions of specific populations with different health programs. Additionally, it was possible to calculate the variability of the health status declared by several people under the same code, enabling the extension of the analysis to 3125 theoretical states.

The hybrid methodology provides several contributions: i) Through individual preferences, it is possible to assign a utility value for each declared health status. ii) Considering the response level given to each dimension in TOPSIS provides the current level of disability; thus, this point reflects the individual characteristics of the person performing the valuation and becomes the essential reference point for obtaining utility values, and the judgments reflect the health and illness experience over time. iii) The results provide useful information, especially for low- and middle-low-income countries, where it is recommended to use simple methods that are not as expensive in primary data collection.

When studying the response to EQ-5D, considering it as a linguistic variable, a better fit to the consistency among preferences and the quantitative value associated with the health status declared through VAS was obtained. However, these findings need to be validated in future research by exploring the incorporation of uncertainty and inaccuracy in the decision-making process by individuals with different initial health statuses. Thus, this analysis must begin by taking into account individual perceptions and then aggregating them by common characteristics, which aims to indicate equity issues in low- and middle-low-income countries. Furthermore, including the regional context could reduce the cognitive bias related to the stated preferences affected by the local health system [67].

In this study, a choice between alternatives does not take place; only the level of criteria assessment (AHP) is taken as a necessary input for calculation in the QALY approach. Thus, combining the paired comparison matrix for a specific person with its health code, which includes each dimension’s levels, must identify the current state of disability. This combination allows us to identify how far the declared health status is from the ideal health status, represented by code 55555 (TOPSIS). Under the proposed methodology, the validity of the results using a predefined set of criteria aims to enable decision-makers to solve conflicting real-world quantitative and/or qualitative multi-criteria problems and to find best-fit alternatives from a set of alternatives in specific, uncertain, fuzzy or risky environments. Although for the current research, the AHP method was selected as the baseline weighting method, the authors recognize that there are other methods such as Paprika and Ca that are more complex and have higher resource requirements that could be used to refine the overall methodology; this could be a topic for future work [68].

.

Reviewer 3 Report

COMMENTS TO AUTHORS:

The study proposes a hybrid methodology that combines the analytic hierarchy process (AHP), the APH by D-numbers extended fuzzy preference relation (D-AHP) and, the fuzzy analytic hierarchy process (F-AHP); with the technique for order of preference by similarity to ideal solution (TOPSIS) to obtain individual and aggregated utility values. This study found that all the tested methods, the ordinal preferences preserved consistency, being the F-AHP-TOPSIS, the best performing method due to the benefits of modeling the response options of the EQ-5D in linguistic terms and providing the lowest Euclidean distance according to the consistency adjustment to the transitivity of preferences. I do have some comments as listed below in the order noted.

Comment 1:

The quality of the data set is very important, especially in a Colombian University-based people. For this reason, please clarify the included criteria and excluded criteria of sample collection in the Study Design section simultaneously.

Comment 2:

Please also conduct the internal verification and external verification in the Proposed Model section to confirm the findings.

Author Response

Dear Reviewer.

We appreciate all your comments and suggestions to improve the paper.

Comment 1: The quality of the data set is very important, especially in a Colombian University-based people. For this reason, please clarify the included criteria and excluded criteria of sample collection in the Study Design section simultaneously.

The suggestions was including in Study Design.

Revised Study Design

This observational, descriptive, and non-experimental study used a non-probabilistic quota sampling from the community of the National University of Colombia at Medellin (Universidad Nacional de Colombia – Sede Medellín). The quotas were segmented into mutually exclusive subgroups of students, professors, and administrative staff based on a specified proportion of the population. In each quota, every individual had an equal and nonzero chance of being included in the study, and the selection of one person in a quota did not affect the inclusion or exclusion of other persons. The sampling selection within each quota was random to minimize selection bias. Hence, the comparison groups differed in their perception and preferences for different health states.

The final sampling reflects the heterogeneity of the population in terms of age, socioeconomic status, and educational level. The inclusion criteria considered active members of the university community who were over 18 years old and agreed to participate in the study. The sample included 301 adults with ages ranging between 18 and 90 years old. The fieldwork period was between June and December of 2017.

Comment 2: Please also conduct the internal verification and external verification in the Proposed Model section to confirm the findings.

The suggestions was including in Study Design and results

Revised Study design

The research constitutes a methodological test, and it is not possible to infer causality or perform an association analysis from it; however, the methodology can be applied to any population or country and includes useful measures for qualitative health status assessment, such as patient preferences with respect to the healthcare process and delivery of health services. Additionally, it provides rigorous research strategies that allow going from individual assessments to the population, without going through an aggregate weighted sum that conceals behavior at the individual level; it can be used with generic surveys at the international level or adapted to specific questionnaires in terms of interventions, territorial contexts or health services.

The survey, applied through personal interviews, included quantitative and qualitative questions in four blocks. In the first block, participants were asked about social and demographic characteristics. In the second block, health dimensions were measured following the printed version of the EQ-5D-5L, authorized by EuroQol under the 27819 code. The third block included the VAS, where the individuals stated their subjective perception of their health status. This information was later used as a reference value to evaluate the proposed hybrid method. Finally, each person created a pairwise comparison matrix with  judgments [48]. Such judgments incorporate intangible and tangible factors in the declared preferences, and the weights of each dimension at the individual and aggregate level were calculated. The dimensions weights are used to form a health index or scale and their internal consistency was test thought Cronbach’s alpha coefficient for each method follow  where n is the number of the health dimension,  is the variance of the th item and  is the variance of the total score formed by summing all dimensions. [49, 50]

We include the next paragraph in results section.

The Cronbach’s alpha coefficient were 0.74, 0.78 and 0.81 for D-AHP-TOPSIS, AHP-TOPSIS and F-AHP-TOPSIS respectively, and we can conclude that the questionnaire has satisfactory internal validity

This manuscript is a resubmission of an earlier submission. The following is a list of the peer review reports and author responses from that submission.

Round 1

Reviewer 1 Report

This paper models the derivation of the health status utility values as a multi-criteria decision-making (MCDM) problem and proposes a hybrid methodology using the AHP, D-AHP, F-AHP and TOPSIS techniques to obtain a value for health status. A survey data was used to verify the validity of the method, and the conclusion that F-AHP-TOPSIS method is the best optimal is obtained.

Although this paper did some work, in my view it is not sufficient for IJERPH. The paper is mainly the application of the existing decision-making method, such as AHP, D-AHP, F-AHP and TOPSIS, and an obvious question is the contribution of this paper is not enough for a publication.

The presentation of the paper should be improved.

1.       When referring to the quality-adjusted life years (QALY), the review of current research result is insufficient. Some more relevant important literature about Health Status Utility should be mentioned, especially, more recently relevant documents should be added. This greatly reduces the quality of this paper, and makes me feel that this research lacks significance and necessity.

2.       It would improve the paper if the statistical results for health surveys would be described more intuitive. In the section of Results and Discussion, the descriptive statistics of the individual surveyed would be readable if represented by charts and tables. Some description in the body do not match the information of the corresponding chart, for instance, line 316-371“At the health states checked (57 out of 3125), 80% of them concentrated in fourteen codes (see Table 4)”, I find noting in Table 4, so a bit more care would be welcome there.

3.       This paper says nothing about the robustness of the results, which reduced the reliability and credibility of the conclusion.

Reviewer 2 Report

Thank you for offering the opportunity to review the manuscript "Hybrid methodology to improve health status utility values derivation using EQ-5D-5L and advanced Multi Criteria techniques". In this paper the authors address the problem that value sets for instruments like EQ-5D reflect social values (the average of a group of people) whilst for certain medical decisions you potentially are more interested to understand how that specific person values his or her current health or possible health outcomes. While the VAS score reflects value for current health, it does not tell us how possible health outcomes would be appreciated. The paper therefore introduces a method that allows an individual value function to be constructed on the basis of attribute weights. I understand this problem and find it worth striving for a solution and I hoped the paper would present a way forward. 

Unfortunately, there are many issues with the paper. My major concerns are with: 

the paper lacks a clear focus on  the problem at hand and for example also describes the major developments in the field of quality of life valuation. A lot of the concepts introduced are not defined, loosely defined, and sometimes used in the wrong way. This makes it extremely difficult to follow the logic of the paper.  the description of the problem addressed lacks clarity (i hope i got right in my summary above) and no theoretical framework is provided to explain why the proposed solution would be adequate to fox the problem. Furthermore the proposed solution is complex and involves many layers of mathematical analyses. The mathematics are not properly explained and again difficult to follow.  The proposed solution departs from very crude preference data, without any justification for using data of that  type and all their inherent limitations. The authors establish ratings of the 5 attributes based on pairwise comparisons of two attributes at the time. There is no information about preferences for severity levels. It is unclear why this is assumed to produce optimal results in view of the availability of alternative strategies that provide richer data about an individual's utility function (e.g. the Paprika method) 

While I could be convinced that the proposed methods have value, the current paper fails to make this clear. If the authors are keen to pursue publication of their idea I recommend that they reconsider their analytical approach and see where it can be simplified, revisited the decision whether the data it is based on is any good, and write a paper that excludes much of the contextual materials story lines and instead present the solution in context of the problem it can resolve (medical decision making based on collected outcomes, e.g. ROM data)